# Medicated Meloxicam Pellets Reduce Some Indicators of Pain in Disbudded Dairy Calves

**DOI:** 10.3390/ani15111641

**Published:** 2025-06-03

**Authors:** Tiarna Scerri, Sabrina Lomax, Peter Thomson, Benjamin Kimble, Peter White, Merran Govendir, Cameron Clark, Dominique Van der Saag

**Affiliations:** 1School of Life and Environmental Sciences, The University of Sydney, Sydney, NSW 2006, Australia; tsce0635@uni.sydney.edu.au (T.S.);; 2Sydney School of Veterinary Science, The University of Sydney, Sydney, NSW 2006, Australia; peter.thomson@sydney.edu.au (P.T.);; 3Gulbali Institute, Charles Sturt University, Wagga Wagga, NSW 2678, Australia; camclark@csu.edu.au

**Keywords:** behaviour, calf, disbudding, medicated feed, meloxicam, pain

## Abstract

Disbudding is the removal of a calf’s horn buds before they fuse to the skull, inducing inflammation at the horn sites and associated pain and discomfort that is detrimental to calf welfare. Current conventional practice includes the administration of a meloxicam injection to ease inflammation. However, this treatment only lasts up to 44 h following administration. The pain and discomfort associated with disbudding is suggested to last for up to two weeks. Recognising the need for longer-lasting pain relief, this experiment tested a novel treatment whereby grain-based pellets formulated with meloxicam were fed to disbudded calves over a 7-day period. In comparison to a conventional meloxicam injection, the medicated meloxicam pellet treatment resulted in less inflammation across the 7-day feeding period. Calves treated with the medicated pellets also exhibited less pain-specific behaviours and more positive social-specific behaviours during and beyond the feeding period. Having concluded the beneficial value of this treatment, further work will focus on reproducing results and optimising treatment practicality.

## 1. Introduction

Disbudding is the surgical removal of a calf’s budding horns before they grow and fuse to the frontal skull bone [1]. Generally performed at 4–8 weeks of age [2], this routine husbandry procedure aids management of individual animal safety as well as collective herd and handler safety [3,4].

This procedure is most commonly executed by cauterisation, where a cautery iron is used to burn off existing horn tissue and prevent further horn growth [1,5]. In creating open wounds, this method causes tissue inflammation and localised nociception [6], leading to increased horn site hyperalgesia [7,8], greater calf stress responses [9,10] and behavioural indicators of pain and discomfort [3,9,11]. Long-term physical effects can include permanent horn site desensitisation [6] and chronic discomfort [12]. Changes in affective state, as recently recognised but nevertheless significant pain indicators [5], have also been recorded as a result of disbudding. Such changes include changed motivations [13,14], social behaviours [14] and biases [5], all of which affect calf affective state and early development to a context-dependent degree. Emerging research reveals that impaired welfare status persists throughout the tissue repair process for up to 2 weeks post-disbudding [15]. Therefore, whilst perhaps justified in the broader interests of production safety [6], disbudding is a prominent welfare concern [16,17,18].

Historically, research has investigated the significance of sedation [19], disbudding technique [20,21] and anaesthetic type [3,11] in ameliorating disbudding pain. Such research has notably improved modern practice, which now commonly includes the administration of a local anaesthetic and sedative during the cautery disbudding procedure [22]. Previous research has also demonstrated the efficacy of the extra administration of nonsteroidal anti-inflammatory drugs (NSAIDs) such as meloxicam [8,23,24]. These compounds inhibit the cyclooxygenase-2 (COX-2) enzyme that initiates wound inflammation, thereby providing analgesic relief around the calf’s horn sites [24,25]. Notably, a single subcutaneous meloxicam injection (0.5 mg/kg) prior to disbudding has been found to effectively reduce behavioural and physiological pain indicators for up to ~44 h following treatment [23,25]. Given this positive welfare outcome, this single-dose method is becoming increasingly conventional.

The implementation of such pain relief protocols effectively manages the short-term pain experienced by calves during and following disbudding for up to 44 h. Yet, there has been minimal investigation into combatting the residual discomfort and consequent detrimental effects associated with the calf’s longer-term tissue repair process and associated pain [26,27].

Cognisant of the need to address this gap, Wilson et al. [28] proposed a novel sustained-release meloxicam pellet formulation and demonstrated that feeding these medicated meloxicam pellets to calves for 7 days resulted in prolonged plasma concentrations of meloxicam compared to a single subcutaneous dose of the drug. In providing sustained rather than single-dose NSAID treatment to calves, the related literature suggests that such a method may promote extended analgesic pain relief [29,30]. Here, we aimed to determine the effects of medicated meloxicam pellets (MMP) on the short-term and longer-term behaviour and physiology of disbudded dairy calves. We also aimed to compare these effects to those resulting from treatment with a conventional single-dose meloxicam treatment. We hypothesised that both treatments would be equally effective at managing calf pain in the first ~44 h following treatment. However, we hypothesised that treatment with the MMP was expected to cause calves to exhibit less behavioural and physiological indicators of pain and discomfort following ~44 h.

## 2. Materials and Methods

### 2.1. Animals and Housing

Forty Holstein–Friesian heifer calves (~4–8 weeks old) requiring routine disbudding were sourced from the University of Sydney’s ‘Corstophine’ property, NSW, Australia. The protocol was approved by the University of Sydney Animal Ethics Committee (2020/1780). Calves were kept in groups of 10 in identical grassed housing paddocks with accessible shelter and the capacity to see and interact with calves in neighbouring paddocks. Calves were grouped in treatment groups within each pen to allow for feeding of MMP to one of the four treatment groups. Prior to the experimental period, all calves were group-fed milk twice daily and calf pellets (Vella^®^ Calf Starter, Vella Stock Feeds, Glendenning, NSW, Australia) (1 kg/head) once daily as per normal property practice. Ad libitum access to water and hay was also supplied. These feeding conditions continued throughout the experimental period, although one of the four treatment groups was fed MMP (1 kg/head/day) instead of unmedicated pellets from Days −1 to 6 of the experiment.

### 2.2. Experimental Design and Treatments

The experiment was conducted in two experimental blocks, with 20 calves per block.

There were 10 calves in each treatment group, randomly allocated but balanced in weight and across experimental blocks. Treatment groups were as follows:(1)POS, *n* = 10: Sham disbudding/manual bud palpation;(2)NEG, *n* = 10: Disbudding and no NSAID administered;(3)MET, *n* = 10: Disbudding with conventional meloxicam (Metacam20^®^, Boehringer Ingelheim, North Ryde, NSW, Australia) subcutaneously injected into the neck at a dose rate of 0.5 mg/kg 2 h prior to disbudding;(4)MMP, *n* = 10: Disbudding with provision of medicated meloxicam pellets (MMP) 1 day prior to disbudding, on the day of disbudding, and for 6 days following disbudding. Calves were fed MMP whilst enclosed in individual pens for a 2 h period. During this time, calves had access to water. If the medicated pellets were not entirely consumed after the 2 h period, calves were released from the individual pens and the remaining MMP were left in a shared trough for all calves to access ad libitum.

Within each experimental block, disbudding occurred on Day 0. Data collection occurred on Days −1, 0, 1, 2, 3, 6, 9 and 12 of the experimental periods. On Day −1, the calves were weighed and spray-painted with individual identification numbers on both flanks and the back. Accelerometers were also attached on Day −1 as henceforth described. On all experimental days, calves were quietly moved into the nearby handling yards within their treatment group. Calves were contained within these handling yards for sedation, disbudding and treatment with local anaesthetic. For data collection, each calf was restrained in the head bale of a cattle crush (Stearman^®^, Wichita, KS, USA). Calves in the MET group were also treated with meloxicam whilst restrained in this cattle crush. Calves were returned to the housing paddocks following completion of experimental activities each day and given their daily pellet ration immediately.

### 2.3. Disbudding and Analgesic Treatment

Disbudding was performed on Day 0 by an experienced livestock veterinarian. Ten minutes prior to sham disbudding or disbudding, all calves were sedated with an injection of intravenous xylazine (Xylazil, Ilium, North Ryde, NSW, Australia, 0.04 mg/kg) into the tail vein. Five minutes prior to disbudding, each calf was administered 5 mL of local anaesthetic lignocaine (2% Lignocaine HCl, Ilium, North Ryde, NSW, Australia) as a cornual nerve block at each horn bud. Disbudding was performed via cautery using a gas debudder (Portasol^®^ Gas Debudder, Cordless MK3, Miami, FL, USA). For two calves, the horn buds were at an advanced stage of development, requiring dehorning via the scoop method [31].

Medicated meloxicam pellets were formulated at a dose rate based on the body weight of the heaviest calf in the experimental group (120 kg) and a feed ration of 1 kg pellets/day. Therefore, to achieve an oral dose rate of ~1 mg/kg meloxicam, meloxicam was incorporated into the pellets at a dose rate of 120 mg/kg pellets. This rate was based on calculations performed by Wilson et al. [28]. The MMP were formulated at the University of Sydney’s feed mill using calf pellets (Vella^®^ Calf Starter, Karl Vella Group Ltd., Skelmersdale, UK) and pure powdered meloxicam.

### 2.4. Data Collection

Data collection occurred once in the morning for all experimental days (between 0800 h and 1130 h) except Day 0, where data collection occurred 4 h following treatment (at ~1300 h).

#### 2.4.1. Plasma Meloxicam

Blood samples (10 mL) were drawn from each calf’s jugular vein into lithium heparin vacutainers (BD Vacutainer^®^, LH 170 I.U., Becton Dickinson, Hong Kong, China). These samples were inverted to distribute its anti-coagulant reagent, immediately put on ice and then centrifuged for plasma transfer. The separated plasma was transferred into microcentrifuge tubes and frozen at −20 °C and submitted for analysis of plasma meloxicam concentration. Quantification of plasma meloxicam concentrations was performed with validated HPLC-UV as previously reported [32,33].

#### 2.4.2. Behavioural Observations

Behaviour was continuously recorded using a CCTV system (Swann Smart Security System^®^, Wi-Fi NVM-490, Swann Security, Melbourne, Australia) set up in the housing paddocks. Four trained blind observers conducted instantaneous behavioural observations for each calf, identifiable by their individual spray-painted number, on each experimental day. The observers used an ethogram (Table 1) outlining normal, pain-specific and social behaviours informed by our previous research and the supporting literature [3,4,7,9,25]. Depending on calf visibility, one observation was taken every 15 min for approximately 3 to 3.5 h, resulting in 12–13 observations for each calf per day. 

#### 2.4.3. Accelerometer-Derived Behaviour States

Tri-axial accelerometers (Hobo Pendant^®^ G, Onset, Bourne, MA, USA) were attached to the left-rear hind limb of all calves using sponge, vet wrap and tape so that the x-axis was perpendicular to the ground when the calves were in a standing position. The accelerometers were programmed to start recording on the x- and y-axes every 20 s for 1 day prior to treatment (Day −1) and up until 12 days following treatment. On Day 6, accelerometers were removed and replaced with new accelerometers to allow sufficient storage capacity on Days 6 to 12 of the experiment. Accelerometer data were used to assess the proportion of the day calves spent lying down (as opposed to standing or walking). To validate lying behaviour as categorised by the accelerometers, 1 h of calf video footage from the first experimental block was synchronised with accelerometer data.

#### 2.4.4. Mechanical Nociceptive Threshold Testing

Prior to testing, calves were blindfolded to eliminate responses to visual stimuli [34]. Mechanical nociceptive testing (MNT) was then conducted using an algometer (Wagner FDIX, Force One, Wagner Instruments, Greenwich, CT, USA), which measured the pounds of force per square inch that could be applied to a specific horn site before the calf exhibited a withdrawal response. The algometer was used to apply pressure to site 1 on the outer intact tissue of each disbudding wound (2 cm distance from the wound edge) or horn bud for POS calves and site 2 on the cut skin edge of each disbudding wound or the edge of the horn bud for POS calves. This resulted in a total of four sites tested on each animal at each sampling time point. Horn sites were measured in the following order: right horn site 1, right horn site 2, left horn site 1, left horn site 2. This test was uniformly conducted by two experienced operators at each time point.

#### 2.4.5. Horn Site Temperature

Horn site temperature was taken using a handheld infrared laser thermometer (QM7215, Digitech, Sandy, UT, USA). The distance between the thermometer and the horn site was kept consistent throughout the experiment using a 15 cm zip tie connected to the thermometer. Horn site temperature was taken at the same horn sites as those described for MNT, and sites were tested in the same order as that described for MNT.

### 2.5. Statistical Analysis

#### 2.5.1. Plasma Meloxicam

From the individual plasma meloxicam concentration data, means for both the MET and MMP treatment groups were calculated. See Appendix A, Table A1, for the individual data.

#### 2.5.2. Behavioural Observations

There were a total of 23 behaviour states recorded (presence = 1, absence = 0). For each of these variables where at least 50 observations of the behaviour were recorded (14 behaviour variables), the data were analysed using a logistic regression model with fixed effects of treatment, day and their interaction. Note that initially, logistic mixed models were attempted by inclusion of block and calf within block random effects (using the lme4 package on RStudio© 2020 [35]), but in most cases, estimated variance components were zero or model fitting did not converge, and hence, fixed-effect-only models were used. These were fitted using the glm function in R. Model-based probabilities of the behaviour being expressed were calculated using the emmeans package.

Upon further behavioural visualisation, ‘head–head’ and ‘head–body’ interactions were collated as ‘head interactions’. Bucking (B), aggression (A) and isolation (I) behaviour data was also collated as ‘anti-social’ behaviours.

#### 2.5.3. Accelerometer-Derived Behaviour States

There were 10,746 records of acceleration in the x- and y-directions, and corresponding lying versus standing behaviour observations were used to develop various classification methods. These were based on records of 20 animals. A generalised additive mixed model (GAMM) technique was selected to model the probability of each animal lying as a smooth function of the two acceleration predictor variables. This was achieved using ‘thin-plate splines’ which allow the probability of lying versus standing to vary in a smooth way over the two variables. The form of the model is
log*_e_* (π/(1 − π)) = constant + *s*(*x*,*y*) + Animal
where π is the probability of an animal being classified as lying (with 1 − π being the probability of standing), *s*(*x*,*y*) is a smooth two-dimensional thin-plate spline function of the acceleration measures in the *x*- and *y*-directions and Animal is a random effect to allow for repeated measures in the same animal over time. Model-based probabilities of lying were classified as lying whenever the probability exceeded 0.5 and were otherwise classified as standing. The gamm4 package in RStudio© [35] was used to fit the GAMMs.

Based on the GAMM method for classifying behaviour as lying, the proportion of time each calf was lying over each experimental day was calculated. Note that accelerometers were switched on Day 6, resulting in cows having two sets of accelerometer readings that day (a.m. and p.m.). The proportion data were analysed by fitting a linear mixed model using the lme4 package in R. Fixed effects for the model were treatment, day and the treatment × day interaction. The random effect for the model was calf number. Model-based means were estimated using the emmeans package, and pairwise treatment comparisons within each study day were performed using the cld function of the multcomp package in R. Note that several repeat analyses were conducted after identifying potentially problematic days and calves.

#### 2.5.4. Mechanical Nociceptive Threshold Testing

A linear mixed model was fitted to the algometer (kg force) data, with fixed effects for treatment, day and site together with all two- and three-way interactions. Random effects in the model to allow for clustering were block, calf nested within block and day nested within calf. Because of the positive skew and unstable residual variance, the force data were log-transformed prior to model fitting. The mixed model was fitted using the lme4 package in RStudio© [35] and model-based means calculated using the emmeans package in R. Pairwise comparisons of means was conducted using the cld function in the multcomp package in R.

#### 2.5.5. Horn Site Temperature

As the temperature data recorded at the horn sites may have been affected by the ambient temperature, half-hourly temperature records for Camden Airport, NSW (nearest meteorological station), were sourced from the Australian Government Bureau of Meteorology for the corresponding experimental period. The air temperature on or at the most recent half hour periods was taken as the relevant air temperature. Then, a linear mixed model was fitted to the horn site temperature data with the same form of that for the algometer data, with the addition of a fixed-effect term for air temperature. No data transformation was required. To allow for a possible nonlinear effect, air temperature was fitted as a spline term using the splines package in RStudio© [35] together with lme4. Model-based means and pairwise comparisons were made, as in the algometer data analysis.

#### 2.5.6. Association Between Algometer Force and Behaviours

To define an objective measure of pain, the algometer force was used (as an inverse measure), and the best set of behaviours that predict that force was investigated. As there were four algometer measurements taken on a calf at a particular time, these values were averaged using a geometric mean (due to the highly positively skewed distribution mentioned in the previous analysis). Similarly, each of the behaviours (binomial 0/1 data) was averaged over all the observations of the calf on that study day (the proportion of times on the day the behaviour was observed). However, rare behaviours (under 50 occurrences in the data set) were not considered as they would not be reliable or have much utility as predictors. The set of daily averages for the algometer and behaviour data were then match-merged on block, calf and day to produce a data set for the analysis.

A linear mixed model was fitted to the algometer (kg force) data, with a fixed effect for the specific behaviour and random effects for block and for calf nested within block. Because of the positive skew and unstable residual variance, the force data were log-transformed prior to model fitting. Each behaviour was tested separately, and any behaviour with a slight association with the algometer force (*p* < 0.2) was identified. These behaviour variables were then entered into an initial multivariable model, and a process of backward elimination was applied to determine a subset of predictive behaviours. All mixed model fitting was conducted using the lme4 package in RStudio© [35].

#### 2.5.7. Miscellaneous Analyses

A variety of linear and exponential correlation tests as well as paired *t*-tests were performed on datapoints related to blood plasma concentration, horn site temperature and behaviour incidences. These analyses were conducted using RStudio© [35].

## 3. Results

### 3.1. Plasma Meloxicam Concentrations

Mean plasma meloxicam concentration was 50.7% greater in MMP calves than MET calves on Day 0. For MMP calves, there was a steady short-term increase in meloxicam concentrations between Days 0 and 3 before stabilisation between Days 3 and 6 (Figure 1). Despite larger standard errors between some mean datapoints (±0.27–0.41), strong linear correlation between individual calf data confirms this trend’s consistency (r = 0.95, *p* = 0.001). A sharp decay in meloxicam concentration was observed in MMP calves from Day 6 following removal of the pellets. MET calves showed a similar decay from Day 0 following injection. An exponential regression test on these decays (Days 6–12 for MMP calves, Days 0–6 for MET calves) found a strong exponential correlation (r = 0.82, 0.7 < r ≤ 1 for strong exponential correlation). Paired *t*-tests revealed a significant difference between MET and MMP means across all experimental days and no significant difference between the means of both experimental blocks per day.

### 3.2. Horn Site Temperature

Having accounted for the nonlinear impact of ambient temperature on horn site temperature data, Figure 2 displays the statistical interactions of each modelled mean per day.

Horn site temperatures of the three disbudded treatment groups were significantly elevated from Day −1 to Day 0. Despite a uniform increase in horn site temperature means, there was no significant difference between the three treatment groups between Days 0 and 2 (Figure 2). There was a significant difference between all three groups on Day 3. The MMP calves displayed lower mean horn site temperatures (Figure 2). The MMP group had the lowest mean temperature on Day 12 (Figure 2). In comparison, both the NEG and MET treatment data exhibited greater mean horn site temperature on Day 3 than on Day 2. Yet, the NEG treatment mean was significantly higher than the MET treatment mean. The MET treatment exhibited a significantly higher temperature mean on Day 12 (Figure 2). Paired *t*-tests confirm a significant difference between the MET and MMP treatment means (*p* = 0.031).

Right horn sites 1 and 2 had significantly higher mean IR temperatures than left horn sites 1 and 2 (Figure 3). This trend was observed on each experimental day except Day 2. Paired *t*-tests also revealed that both horn sides had significantly greater mean temperatures on site 2 than site 1 (left *p* = 0.0114, right *p* = 0.0239). Statistically, the MMP treatment mean was found to significantly differ on both right horn sites.

### 3.3. Accelerometer-Derived Lying Time

There was a significant treatment × day interaction (*p* < 0.001). However, despite established treatment differences on Day –1 (Appendix A, Table A2), there was no treatment difference between the mean proportion of time spent lying between the treatment groups across Days 0–3 (Appendix A, Table A2). There was no significant long-term difference from Days 6–12 (Appendix A, Table A2).

### 3.4. Mechanical Nociceptive Threshold

Treatment groups did not significantly differ in mean mechanical nociceptive threshold (MNT) on Day −1 (Appendix A, Table A3). On Days 0 and 1, the three disbudded treatment groups exhibited a reduction in MNT. The MMP treatment had the lowest MNT on both days. However, from Day 2, the three disbudded treatment groups maintained a significantly lower mean MNT than the POS treatment with no significant difference between them. Interestingly, when the data were visualised as a variation from each treatment’s respective baseline MNT on Day −1, there was a lower level of variation for the MMP treatment (Figure 4). These differences were greater for the NEG and MET treatments.

There was a consistently lower algometry score across all three disbudded treatments for both left and right horn sites 2 than the wound site (Appendix A, Figure A1). This difference was significant across all days of the experimental period.

### 3.5. Behaviour

There were several statistically treatment × day interactions for the 14 analysed behaviour traits (Appendix A Table A4). Ear flicking (EF) and head shaking (HS) were the most frequently observed behaviours across the experimental period. Ear flicking occurred at low levels prior to the disbudding procedure for all treatment groups, followed by a significant increase in all treatments but the MET group on Day 0 (Figure 5a). There was no difference between MET and MMP treatments across the experimental period (*p* = 0.28).

There was an increase in HS behaviour from Days −1 to 0 across the disbudded treatment groups (Figure 5b). There were insufficient observations of tripping (T), rearing (R) and pawing (P) behaviours.

In terms of general behaviour, the MET treatment exhibited the largest decline in E behaviour, and the MMP treatment retained the highest level of E behaviour (Appendix A, Figure A2). Yet, paired *t*-tests show that this difference was not sufficiently significant (*p* = 0.387). Day 12 featured minimal E behaviour from all treatment groups. There were no discernible trends in foot stamping (FS) and tail flicking (TF) data nor in proportions of standing and lying statuses.

The POS calves performed more head interaction behaviours from Days 0–6, while the MET and NEG groups displayed the lowest levels of head interaction behaviour from Days 0–6 (Figure 6a). MMP calves exhibited more head interaction behaviours than the MET and NEG groups (Figure 6a). There was a significant difference between the MET and MMP treatments (*p* = 0.014). There was a low occurrence of running behaviour, which was almost exclusively exhibited by the POS and NEG treatment groups. POS calves displayed low occurrence of ‘anti-social’ behaviours across the experimental period. Figure 6c shows a similarly minimal level of anti-social behaviour from each disbudded treatment group on Days 0 and 1. However, both the MET and MMP treatments exhibited an increase in anti-social behaviour on Day 2. Although the MET treatment maintained the incidence of anti-social behaviours across Days 3, 6 and 9, the MMP treatment calves exhibited an incidence reduction on Day 3 and no anti-social behaviours on Days 6 and 9 (Figure 6c). No calves exhibited anti-social behaviours within Day 12 observations. The difference between the MET and MMP treatments was not significant (*p* = 0.119).

## 4. Discussion

This experiment is one of the first to comprehensively investigate the longer-term effects of disbudding conducted with both anaesthetic and analgesic treatments. In itself, this achievement is an important step towards understanding the prolonged welfare implications of disbudding under current industry practice. However, this study is the first study to investigate the efficacy of sustained analgesic treatment via a pellet formulation.

### 4.1. Sustained Meloxicam Concentrations in MMP Treatment

The pellet formulation and feeding regime applied to the MMP treatment calves successfully resulted in sustained plasma meloxicam concentrations throughout the 7-day feeding period. As a replication of Wilson et al. [28], this result reinforces the effectiveness of the treatment method at sustained meloxicam delivery. Although there is no current consensus on the minimum effective plasma concentration for meloxicam treatment in dairy calves [36], the MMP treatment calves featured higher mean plasma meloxicam concentration levels on Days 0–6 than the assumably effective dose for the MET group on Day 0. That level is assumed, but not guaranteed, to be sufficient at catalysing a pharmacological response. Expectedly, there was higher standard error and thus higher individual variability in the MMP group than the MET group. This variability can be attributed to inherent differences in calf appetite and thus their voluntary pellet intake. The fact that the MMP treatment exhibited a 50.7% higher plasma meloxicam concentration is likely due to the shorter half-life of the subcutaneous injection at Day 0 [37].

### 4.2. Less Prolonged Wound Inflammation in MMP Treatment

With Day −1 having established no difference in horn site temperature means between the disbudded treatment groups, the MMP treatment exhibited lower horn site temperature means on all post-disbudding experimental days except Day 9. In the short term (Days 0–2), the treatment temperature mean was not significant from the other disbudded treatment groups. It was hypothesised that with both MET and MMP treatment groups having active meloxicam in their system, these treatment groups were not likely to differ in the short term. However, the fact that there was no significant difference between these treatments and the NEG treatment evokes the suggestion that there was either a delayed response in calf inflammation response or meloxicam anti-inflammatory activation. In the long term (Days 3–12), the difference between temperature means per disbudded treatment was significant on all days except Day 9. Considering that the MMP treatment had meloxicam dosage on Days 3 and 6, it is rational that the meloxicam would actively have an anti-inflammatory effect and thereby minimise treatment increase in horn site temperature mean and consequent wound inflammation. Closer day-by-day results could have provided more clarity into the reason for Day 9 outlier data, e.g., an inflammation resurgence. Nevertheless, the fact that a significant difference between means continued past Day 6 onto Day 12 suggests that sustained MMP treatment may also play a role in mediating the inflammation response past the 7-day feeding period. However, due to the Day 9 outlier, such a conclusion would require further experimentation. To summarise, the MMP treatment calves exhibited a significantly lower long-term level of wound inflammation than the MET treatment.

The most rational explanation for the unexpectedly higher POS temperature means is horn bud heat absorption from sunlight. Although the data were modelled to account for ambient temperature per data collection time point per experimental day, the calves’ intact horn buds were still able to absorb heat from the sun while they were mustered in the unshaded experimental area. This heat absorption—likely exacerbated by the fact that the POS calves were consistently the last test group—is the strongest explanation for their consistently higher temperature means. Given the suggested role of fully developed horns in heat dissipation [38], an alternative explanation is that the POS calves’ horn buds were a conduit for heat dissipation for their whole bodies. However, it is unknown whether calf horn buds have developed the necessary blood vessel infrastructure for such heat dissipation and to the extent observed. Either way, without horn buds, all disbudded treatment groups were not subjected to this confounding variable and thus their data were not as affected. Furthermore, the fact that the right horn site means were significantly higher than the left horn site means is likely due to direct sunlight from the east increasing the infrared temperature on the calf’s right side. This phenomenon has been observed in other trials using infrared thermometers [39,40]. Although this trend has consistently affected all treatments and therefore does not affect the validity of differences between the disbudded treatment groups, sunlight intensity has nevertheless been shown to be an uncontrolled variable that should be minimised. In future studies, infrared temperature data collection should be conducted in a fully shaded area. It may also be valuable to take multiple temperature measurements per site for verification.

Although meloxicam as an NSAID was expected to have a clear impact on wound inflammation, it was also expected that a reduction in the inflammation of wound tissue would affect calf nociceptive response [41]. However, aside from some differences on Day 0 and 1, the MNT means of each disbudded treatment group were insignificantly different. Yet, upon measurement of the treatment difference in means as a variation from the Day −1 baseline MNT mean, the MMP group consistently presented the least MNT decline whilst the MET and NEG groups comparably exhibited a higher decline. This perspective indicates that there was some, albeit insignificant, treatment effect between the disbudded treatment groups that was hindered by a lack of data strength. It is likely that the lack of significant difference between the disbudded treatments stems from a higher level of individual calf MNT variability, as evidenced by the medium level of standard error. Indeed, genetics are hypothesised to play a role in the extent of nociceptive activation and sensitivity [42,43]. This threshold variability suggests that a higher number of biological replicates may be needed in further trials to produce more significant data trends. Nevertheless, the lack of result highlights the distinction between the inflammation and nociception pathways [44].

### 4.3. Higher Sociality in MMP Treatment

The most interesting results from the behavioural observations were within the social-specific behaviour category. Head interaction behaviour, as a subset of play behaviour, can generally be attributed to a positive affective state in which a calf is curious and/or willing to form a relationship with another calf [45]. Therefore, a decline in head interaction behaviour may be indicative of an impaired affective state prompted by pain or discomfort. Viewing the POS treatment as a baseline of normal head interaction behaviour, the MMP treatment is, importantly, the most comparable with the POS treatment across the majority of the experimental period. Furthermore, there is a clear significant difference between the higher head interaction behaviours of the MMP and lower incidences of such behaviours in the MET calves despite inconsistent Day 9 and 12 results. Such maintained social behaviour for the duration of active analgesic treatment is a replicated finding [22]. Therefore, this overarchingly greater willingness from the MMP group to interact with their head area, the key source of their pain and discomfort, strongly suggests that their affectivity is sufficiently positive to maintain normal indicators of sociality. Despite some treatment, day and treatment × day interaction results, the MMP vs. MET treatment differences were not replicated within individual analysis of head–head interaction (HHI) or head–body interaction (HBI) behaviours. Although this outcome could be an instance where a greater number of replicates solidifies trend validity, it is more likely a product of the instantaneous sampling method incorporated within the experimental design. Although instantaneous sampling beneficially allows a balanced snapshot of calf behaviour across each day, the limited time period also means that quick, unpredictable calf interactions are likely to be missed more often than observed. Thus, future behavioural studies would likely gain a more in-depth understanding from incorporating both focal and instantaneous behavioural sampling across each experimental day. Nevertheless, the MMP treatment still showed significantly fewer indicators of negative affectivity. This behavioural finding is also reinforced by the collective behavioural index. The MMP treatment exhibited significantly different probability scores from the MET group on all experimental days except Day −1. Consequently, in viewing these changes as a result of disbudding treatment, the data show that the MMP treatment calves were consistently less likely to exhibit symptoms of pain and discomfort. The poignancy of this discrepancy is clearest on Day 3, with the MET group calculated to be greatly more likely to exhibit a more severe behavioural score as an indication of greater pain, discomfort and general negative affectivity. Admittedly, some of the POS and NEG predictions are skewed, likely as a result of insignificant general behavioural changes such as normal lying (NL) being overly represented in the algorithm. Nevertheless, the MMP vs. MET contrast in estimated affectivity is clearly replicated on both a short-term and long-term collective scale. Overall, the head interaction behavioural results show a significantly more positive, long-term affective state in MMP treatment calves over MET calves. The results also demonstrate the future scope for monitoring social-specific behaviours and ultimately affectivity as a nexus for measuring calf pain and discomfort.

Facilitating the notion that different calves of different dispositions would display a lack of social receptivity via different behaviours [46], ‘anti-social’ behaviours included not only agonistic behaviours such as bucking (B) and aggression (A) but also isolation (I). Such anti-social behaviours were also hypothesised to give insight into the calves’ affective state, with higher levels of anti-social behaviour potentially indicating a negative affective state. Despite a low level of anti-social behaviours immediately post-procedure, all disbudded treatment groups featured a marked increase in anti-social behaviour from Days 2 to 9 (with only a Day 2 NEG outlier due to low observation availability). The strongest explanation for this result is frustration, in that the experimental calves were experiencing some level of discomfort that was not ceding over time and therefore their frustration was increased. This frustration, in worsening their affective state, thereby led to higher incidences of anti-social behaviour. The data showed the MMP treatment to display consistently lower incidences of anti-social behaviour, whereas MET calves showed long-term levels of anti-sociality comparable with the NEG group. However, similar to the algometry data, these trends were not statistically significant. Further experimentation with a higher number of biological replicates per treatment may allow these trends, and the largely unexamined concept of calf frustration, to be more seriously examined.

The additional insignificance of pain-specific behavioural data was a surprising result. Even with strong treatment, day and treatment × day interactions, there were insufficient statistical differences between the MMP and MET treatments. This lack of result is likely attributable to gaps in observation availability for the MMP treatment on Day 3 and the MET treatment on Day 6, ultimately contributing to outlying datapoints for each treatment on their respective day. It is also difficult to establish a behavioural baseline for each treatment from one pre-disbudding day of observations, especially when the incidence of certain pain-specific behaviours such as ear flicking (EF) can be confounded by flies or other external stimuli. Although observers cannot control when a calf disappears from view behind a feeder, further trials can be refined by the inclusion of several pre-disbudding days of observation to establish a more consistent behavioural baseline for treatment comparison.

Apart from the expected decline in eating (E) behaviour post-procedure on Day 0, the fact that there were no discernible treatment trends in E behaviour across the experimental period positively shows that the pellet treatment, as an addition to the MMP calves’ diet, had minimal impact on their normal foraging behaviour and ingestion of essential nutrients.

### 4.4. Need for Sustained Disbudding Treatment

Another key insight from this experiment is that indicators of disbudding pain and discomfort persisted throughout the experimental period. Mean horn site temperature for the disbudded treatment groups was still elevated by a significant margin from Day −1 on Day 12, which demonstrates an ongoing level of wound inflammation. Algometry scores were also significantly lower than the POS treatment on Day 12, indicative of abnormal nociception. There were still a relatively high incidence of pain-specific behaviours across Days 6–12. As already discussed, incidences of social behaviour remained relatively low—notwithstanding data gaps on Days 6 and 9—and the long-term increase in anti-social behaviours particularly for the NEG and MET groups suggest a building of frustration for disbudded calves as these detrimental symptoms persisted. Overall, these results add further strength to the existing literature argument [26] that cautery disbudding procedure—even with industry-approved anaesthesia and analgesic treatment—significantly contributes to decreased calf health and welfare status for at least 12 days post-procedure. Accordingly, assuming the continuation of cautery disbudding as the industry standard in Australia, such evidence reinforces the clear need for sustained post-disbudding treatment.

### 4.5. Minor Insights

The general insignificance of accelerometery and standing–lying behaviour was not unexpected. Although some studies have found short-term changes in post-disbudding calf lying behaviours [47,48], there are an equal number of studies that found no change [49,50]. To the authors’ knowledge, no studies have thoroughly examined standing–lying behaviours for disbudded calves long-term. This study’s lack of results cannot be attributed to an insufficient replicate number, when Sutherland et al. [47] yielded significant post-disbudding differences in lying behaviour—as well as treatment differences—with the same number of calves per treatment. Instead, it is possible that the literature’s inconsistent record of disbudding impact on standing–lying behaviours is due to unknown external variables, e.g., bedding, ambient conditions [51,52], also acting upon experimental calves to variable degrees. Indeed, this reasoning would explain the significant day × treatment difference noted in this study’s lying behaviour observations but lack of discernible trends in all other analyses. Such a proposition mainly supports the necessity of establishing longer-term pre-treatment observations and monitoring of known impactful variables like ambient temperature. The literature reference [11] also places emphasis on increased standing–lying transitions as an indicator of ‘restlessness’ and thus a negative affective state. Such behaviour was insufficiently observed in this study, likely due to the instantaneous mode of behavioural sampling within the experimental design. It is again suggested that future studies include a blend of instantaneous and focal behavioural sampling to fully capture the extent of calf behavioural change.

It was similarly rational that disbudded calves returned significantly lower algometry scores on horn site 2 as the horn site directly on the wound and therefore the most inflamed and sensitive to nociception. Nevertheless, this result gives credence to the importance of documenting and maintaining consistency with exact horn bud locations when performing MNT tests.

Running (Rn) behaviour was classed as a positive social-specific or play behaviour in the previous literature [53,54]. It is fair to assume that the somewhat consistent Rn behaviour from the POS group is indicative of positive social interactions. However, the NEG treatment exhibited the highest incidence of Rn behaviour out of all treatment groups by a significant margin on Day 0 alone. Considering these calves were not treated with analgesia and expectedly feeling the most amount of pain and discomfort, it is more likely that this Rn behaviour was indicative of a negative affective state. Potentially, it may even suggest a ‘fight-or-flight’ stress response as a result of pain from an unknown stimulus [44]. This result firmly suggests that further experiments involving the observation of running behaviour should view the behaviour as context-dependent rather than assumedly positive. In the context of immediate post-disbudding behaviour, it potentially should be alternatively viewed as a negative pain-specific behaviour. Further observations of post-disbudding Rn behaviour are needed to solidify this conclusion.

It is also noted that variable pellet intake within the MMP treatment may have contributed to not only variability of meloxicam plasma concentration levels but also further individual results. With a relatively small treatment size of ten calves, such variability—without exact intake measurements—has served as a minor limitation on the depth of some conclusions. Reflecting on this insight, it is recommended that future studies either measure individual calf pellet intake per day or increase treatment replications.

### 4.6. Welfare Value Judgment

Pursuant to the main experimental aim, a comparison of the MET and MMP treatments is warranted. In the short term, 48 h post-disbudding, there was no significant difference between MET and MMP horn site temperature means. Given the lack of Day −1 difference, this result indicates fairly similar levels of wound inflammation across Days 0–2. Despite some statistically significant differences between differences in social-specific behaviours indicative of discrepancies in calf affective state, there is relatively scarce evidence of either treatment conferring a welfare advantage over the other. Indeed, this scarcity is despite the MMP group having a significantly higher mean meloxicam concentration across Days 0–2. As prefaced in Section 4.2, further inquiries should be made into the point of meloxicam activation per dosage for both treatments. Nevertheless, the consequent conclusion that both treatments were similarly effective at mitigating disbudding pain and discomfort meets the initial hypothesis. From a longer-term scope over the 12-day experimental period, the MMP treatment not only presented significantly lower levels of horn site temperature means and thus wound inflammation but also exhibited significantly higher incidences of sociality than the MET treatment. These results provide weight to the contention that the MMP treatment experienced less inflammation-based pain and discomfort, ultimately resulting in a more positive affective state and greater welfare status than the MET calves. In finding the MMP treatment as the better longer-term treatment method, the second hypothesis is fulfilled. Combining both results, it is concluded that the novel MMP treatment confers a welfare advantage over the conventional injection. Given the amount of behavioural and physiological indicators monitored in this trial, this conclusion should have been strengthened by a broader range of statistically significant evidence. Nevertheless, this paper serves as strong preliminary support for this sustained NSAID administration option.

### 4.7. Future Directions

Having established the welfare advantage conferred by MMP treatment, there are a range of future research directions.

However, experimental replications first need to be conducted to verify the reproducibility and thus reliability of these findings and the consequently derived conclusions. The suggested method refinements, e.g., shaded experimental areas and longer pre-disbudding observations, should be applied to these replications. Measurement of other physiological and behavioural variables indicative of welfare status, e.g., handling stress, may also be viable experimental improvements.

Given that this study has reinforced the long-term nature of disbudding impact on calf health and welfare status, there is a question of whether the 7-day feeding period and consequent analgesic relief is sufficiently prolonged for the level of pain and discomfort experienced by disbudded calves. Further research should therefore focus on determining a long-term dosage schedule that best mitigates disbudding pain and discomfort without causing any adverse effects indicative of overly prolonged NSAID administration [55].

It is also worth considering whether MMP treatment protocol is the most practically viable treatment option for dairy farmers. Although already fairly cheap and time-efficient as a feed rather than injection option, a survey may find that the effort of weighing out pellets per calf is perceived to outweigh the treatment’s welfare value. This would necessitate investigation into less individualised administration, e.g., molasses licks [56] or bulk feeding with a lower meloxicam pellet concentration to prevent overdose.

Optimising the efficacy and practicality of this sustained pellet treatment will ultimately strengthen the case for its commercialisation and consequent availability for on-farm use. However, such application is not just limited to disbudding pain management. With further research, MMP treatment could also be applied to more effectively mediate similarly painful branding and castration procedures as well as injuries and diseases that contribute to lameness [57]. For this research direction, further work would need to be dedicated to assessing the most effective formulation and dosage for calves of different sexes and ages.

As such, MMP treatment holds potential as a welfare improvement not only within disbudding practice but across a variety of husbandry procedures and conditions.

## 5. Conclusions

This experiment aimed to test the efficacy of a novel meloxicam pellet treatment at minimising and managing pain and discomfort in disbudded calves in comparison to the conventional meloxicam injection. A variety of behavioural and physiological indicators were monitored on a short-term and long-term basis to achieve this goal. It was found that disbudded calves treated with medicated meloxicam pellets experienced significantly lower wound inflammation and thus inflammation-based pain and discomfort across the long-term experimental period. They also expressed significantly higher sociality than other disbudded treatment groups and the least incidences of anti-social behaviours, indicative of a more positive affective state. In contrast, calves treated with the conventional single-dose meloxicam injection performed equally in terms of short-term pain management and poorer with regard to long-term pain management. These results culminate to allow the important conclusion that the medicated meloxicam pellet treatment yielded a long-term welfare advantage over the conventional treatment. Furthermore, the general persistence of pain and discomfort indicators across the 12-day experimental period reinforces the need for sustained disbudding treatment to become the new industry standard. Future research will focus on determining best on-farm practice and, ultimately, making medicated meloxicam pellets an available analgesic treatment for improved welfare outcomes across the dairy industry.

## Figures and Tables

**Figure 1 animals-15-01641-f001:**
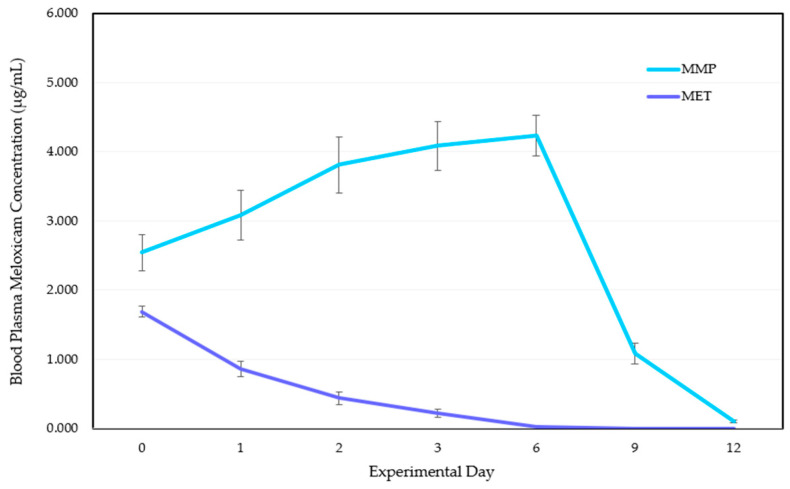
Mean blood plasma meloxicam concentrations (μg/mL) ±1 SE on each experimental day (0, 1, 2, 3, 6, 9 and 12) for calves in treatment groups MET and MMP. (MET = calves treated with a subcutaneous meloxicam injection at 0.5 mg/kg on Day 0 (*n* = 10); MMP = calves treated with medicated meloxicam pellets at a dose rate of ~1 mg/kg on Days −1, 0, 1, 2, 3 and 6 (*n* = 10).) The lower limit of quantification (LLOQ) is 0.10 µg/mL.

**Figure 2 animals-15-01641-f002:**
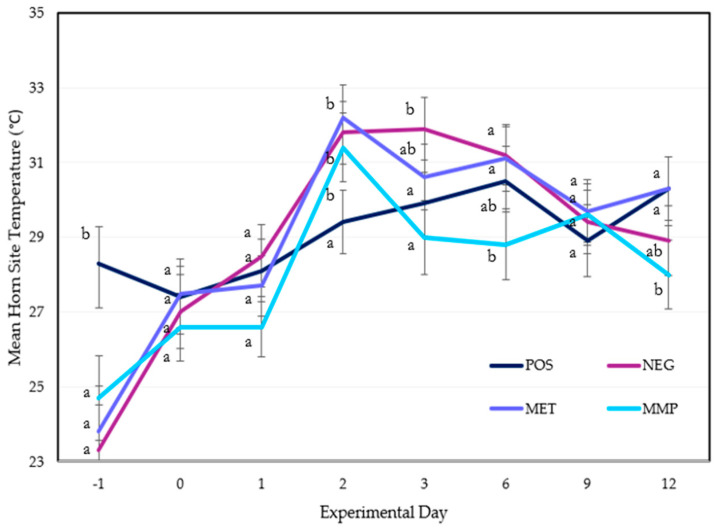
Model-based mean temperatures (°C) ±1 SE for each treatment (POS = positive control (*n* = 10); NEG = disbudding (*n* = 10); MET = disbudding with subcutaneous meloxicam (Metacam20^®^, Boehringer Ingelheim) (*n* = 10); MMP = disbudding with meloxicam pellets) across experimental days (*n* = 10). Treatment groups sharing a similar superscript (^a^, ^b^) do not differ significantly.

**Figure 3 animals-15-01641-f003:**
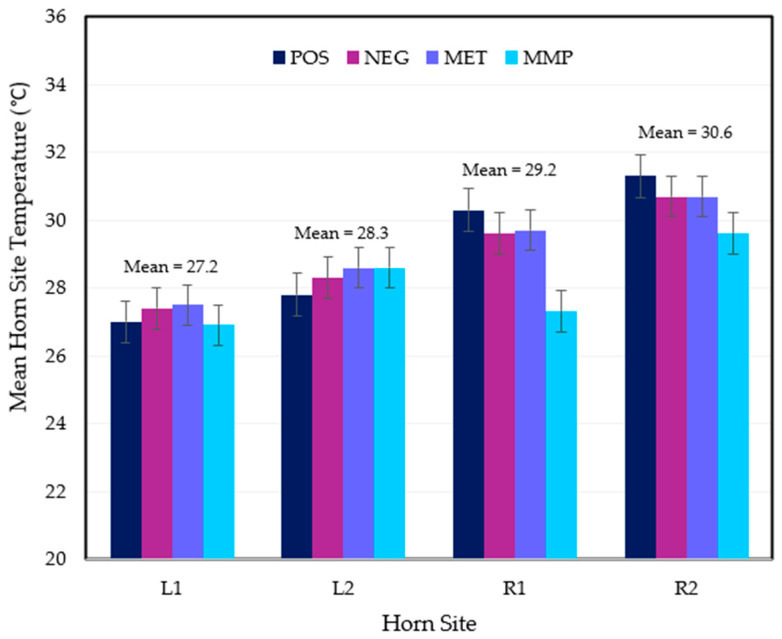
Model-based means of horn site temperature (°C) ± 1 SE for each treatment group (POS = positive control (n = 10); NEG = disbudding (n = 10); MET = disbudding with subcutaneous meloxicam (Metacam20^®^, Boehringer Ingelheim) (n = 10); MMP = disbudding with meloxicam pellets (n = 10)) for left and right horn sites 1 and 2 (L1, L2, R1, R2). Means across treatment groups per horn site are also featured.

**Figure 4 animals-15-01641-f004:**
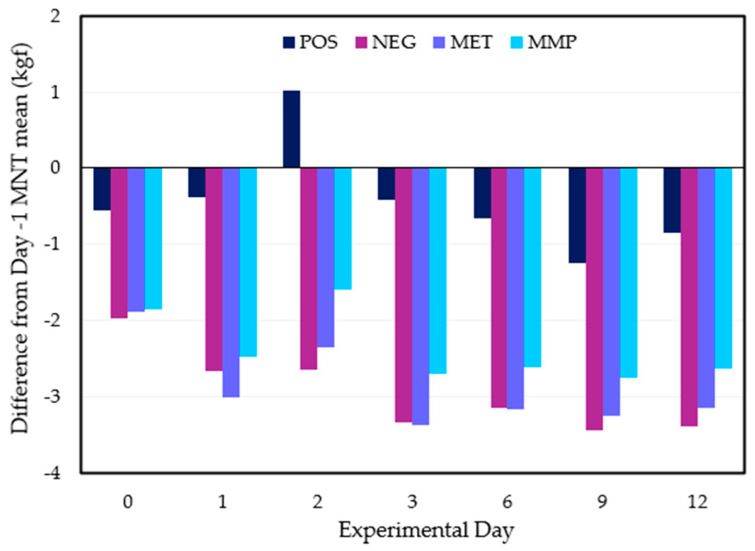
Differences in mean mechanical nociceptive threshold (MNT) from Day −1 MNT mean for each treatment group (POS = positive control (*n* = 10); NEG = disbudding (*n* = 10); MET = disbudding with subcutaneous meloxicam (Metacam20^®^, Boehringer Ingelheim) (*n* = 10); MMP = disbudding with meloxicam pellets (*n* = 10)).

**Figure 5 animals-15-01641-f005:**
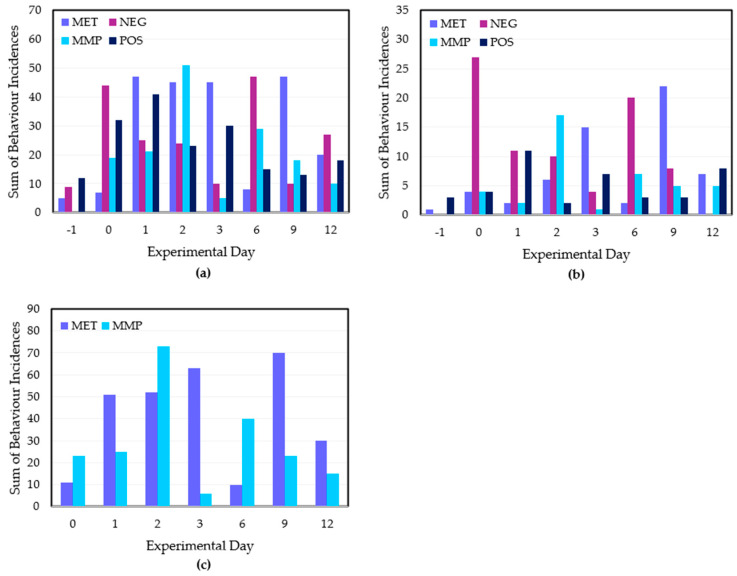
(**a**) Incidences of ear flicking behaviours per treatment group across the experimental period. (**b**) Incidences of head shaking behaviours per treatment group across the experimental period. (**c**) Incidences of all pain-specific behaviours for the MET and MMP treatments across the experimental period. Pain-specific behaviours include ear flicking, head shaking, pawing, tripping and rearing. The treatment groups are POS = positive control (*n* = 10); NEG = disbudding (*n* = 10); MET = disbudding with subcutaneous meloxicam (Metacam20^®^, Boehringer Ingelheim) (*n* = 10); MMP = disbudding with meloxicam pellets (*n* = 10).

**Figure 6 animals-15-01641-f006:**
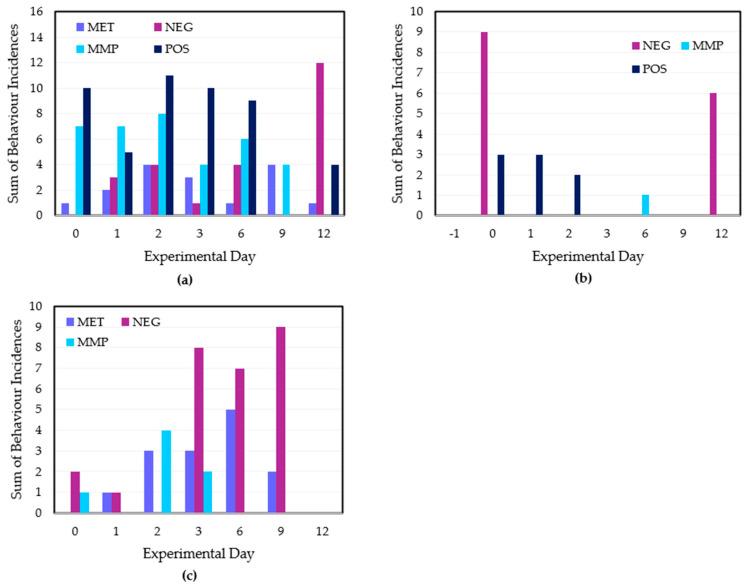
(**a**) Incidences of head interaction behaviours for all treatment groups across the experimental period. (**b**) Incidences of running behaviour for all treatment groups across the experimental period. (**c**) Incidences of anti-social behaviours for the three disbudded treatment groups across the experimental period. Head interaction behaviours include both head–head and head–body interactions. Anti-social behaviours include aggression, bucking and isolation behaviours. The treatment groups are POS = positive control (*n* = 10); NEG = disbudding (*n* = 10); MET = disbudding with subcutaneous meloxicam (Metacam20^®^, Boehringer Ingelheim) (*n* = 10); MMP = disbudding with meloxicam pellets (*n* = 10).

**Table 1 animals-15-01641-t001:** Ethogram of recorded behaviours.

BEHAVIOUR	DEFINITION
**Pain-Specific**	
Ear flicking (EF)	Calf rapidly moves one or both ears independent of head movement or external stimuli, e.g., flies.
Head shaking (HS)	Calf rapidly moves whole head side to side or up and down independent of external stimuli.
Rearing (R)	Transfer of calf bodyweight to hind legs with both fore legs raised simultaneously.
Tripping (T)	Rapid alternate lifting of two or more fore or hind legs independent of other motive, e.g., regaining balance. Event noted each time calf starts lifting legs from having all four on the ground.
Pawing (P)	Calf lifts hind leg and arches neck to scratch or attempt to scratch top of their head with foot.
**General**	
Head rubbing (HR)	Calf rubs head against another object, e.g., pen wall, feeder.
Tail flicking (TF)	Calf rapidly moves tail from side to side ~2–3 times.
Foot stamping (FS)	Calf raises hoof and firmly brings it back down.
Vocalisation (V)	Pronounced vocal noises by calf independent of external stimuli, e.g., vocalisation in response to other calf vocalisation does not count. Signs include constricted sides, extended neck, etc.
Grooming (G)	Calf licks any part of self for more than 1 s.
Drinking (D)	Uptake of water from provided trough/container, where licking the surface does not count.
Eating (E)	Feed uptake from provided container.
Standing/lying transition (S/LT)	Transition of calf from being upright on all four limbs to lying with lower flank in contact with floor. Vice versa applies.
Locomotion (L)	General walking activity around pen, including walking around the feeder and trough.
Standing: normal (NS)	Calf is stationary and upright on all four limbs with relaxed features and head level with topline.
Standing: abnormal (AS)	Calf independently assumes standing position that deviates from normal calf posture for more than 10 s, e.g., extremely raised head. Note nature of abnormal posture.
Lying: normal (NL)	Calf is stationary with lower flank in contact with floor. Limbs may or may not be tucked in under or close to flank.
Lying: abnormal (AL)	Calf independently assumes lying position that deviates from normal calf posture for more than 10 s, e.g., lying on whole side flank with limbs extended. Note the nature of abnormal posture.
**Social**	
Running (Rn)	Any gait faster than a walk, e.g., trot.
Head–head interaction (HHI)	Heads and/or neck of two calves touch for 1+ s. Includes sniffing and licking.
Head–body interaction (HBI)	Any part of the calf’s head contacts part of another calf for 1+ s. Includes sniffing and licking.
Bucking (B)	Calf’s bodyweight shifted from front to back, with both hind hooves lifted off the ground.
Aggression (A)	Interaction with other calves that involves pushing/shoving and attempts to ram another calf with their head.
Isolation (I)	Calf avoids main group independent of external factors.

## Data Availability

The raw data supporting the conclusions of this article will be made available by the authors on request.

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
