# Peer review of "Medicated Meloxicam Pellets Reduce Some Indicators of Pain in Disbudded Dairy Calves"

_animals, 2025, doi:10.3390/ani15111641_

Round 1
Reviewer 1 Report
Comments and Suggestions for Authors
The manuscript titled ‘Medicated meloxicam pellets reduce some indicators of pain in disbudded dairy calves’ by Scerri et al. demonstrates the need for extended analgesic management beyond the first 48h after disbudding and optimizes drug delivery to a larger cohort of animals without need for daily injections by using a medicated diet for pain management.
The paper is written comprehensively and structured well. There are a few major and minor points that should be addressed to improve the translation of the key message as well as the reading experience.
- The data presented in this manuscript is extensive and could be organized better, to prevent the reader from getting lost in the various ways the same data was analyzed. It is strongly recommended to organize several of the figures/tables into a supplemental section or to leave some of them out, especially if the figure/table does not add to the key message of an endpoint. Repetition should be avoided. This pertains to all figures/tables in the manuscript. Examples for this would be:
- 3 Model-based means of horn site temperature – Figure 2 already shows that horn site temperatures were elevated in disbudded groups, and that MMP seemed to achieve a reduction in wound inflammation in the longer-term observation period. The results from specific horn site left/right temperature measurements are only discussed in regard to the POS group and potential reasons for elevated temperatures at the horn sites for this specific group. Fig. 2 seems sufficient to communicate the key message to the reader, and Fig. 3 could be moved to the supplemental section.
- Nociceptive threshold (Table 3, Fig. 4 and 5) – There does not seem a lot of value in analyzing and depicting the same data 3 different ways to later determine that ‘the lack of significant difference between the disbudded treatments stems from a higher level of individual calf MNT variability’. This should be consolidated to one figure or table, with others left out or moved to a supplemental section.
- The simple summary and abstract, which draw in the reader’s attention for a publication both mention that ‘pain and discomfort/ symptoms associated with disbudding is/are suggested to last up to 62 days’, which immediately raises the question why the study was designed for a 7-day treatment and 12-day observation period of calves with analgesics post-disbudding, and not for longer? Suggestion to re-phrase the statement and/or discuss the study design choices. In the introduction it is mentioned that ‘Emerging research reveals that impaired welfare status persists throughout the tissue repair process, which can total up to 62 days’, which gives some more context to the statement from before, but could still raise confusion about the study design. I was unable to verify the source for this (Ref. 15 is given, which is ‘ Meijboom, F.; Stafleu, F. Farming ethics in practice: from freedom to professional moral autonomy for farmers. Agriculture 938 and Human Values 2015, 33, 403-414., which did not contain information on the 62 day statement as far as I could see). In general, the references section seems to be somewhat erroneous, see also below.
- Figure comments
- All figure legends would benefit from the addition of group sizes to the legend.
- 1 – Would benefit from adding the effective therapeutic plasma concentration of meloxicam in calves as a horizontal line
- Discussion comments
- Section 4.1 – It should be stated if the meloxicam plasma concentrations represented the effective dose for calves on all days of testing.
- Section 4.2 – The choice of IR temperature as endpoint/method should be critically discussed for unexpected temperature measurements. The method is known to only provide information on surface temperatures, which can be affected by outside influences, as you discussed. It is also recommended to take at least 3 separate measurements and average them for one timepoint, which I believe wasn’t done here.
- Section 4.6 – Another point that could be added here is that provision of analgesics in form of a medicated diet reduces animal handling stress and prevents needle-shyness in animals, thus improving animal welfare. This becomes especially relevant if provision of analgesia beyond the first 48h after disbudding is pursued.
- References
- At least in the version I am reviewing, the reference section appears to be misaligned – some references seem to be formatted as sub-references, thereby creating number repetitions. Please review and update if found erroneous on your end.
- Minor comments
- Section 2.4.1 – Should state the frequency of blood collections
- Section 2.4.2 – Text reference to ethogram/Table 1 should be added
- Line 48 – Add comma after ‘hyperalgesia’
- Line 288 – Repeat word ‘that’ can be deleted
Author Response
Thank you for your feedback.
Comment: “The data presented in this manuscript is extensive and could be organized better, to prevent the reader from getting lost in the various ways the same data was analyzed. It is strongly recommended to organize several of the figures/tables into a supplemental section or to leave some of them out, especially if the figure/table does not add to the key message of an endpoint. Repetition should be avoided. This pertains to all figures/tables in the manuscript. Examples for this would be:
3 Model-based means of horn site temperature – Figure 2 already shows that horn site temperatures were elevated in disbudded groups, and that MMP seemed to achieve a reduction in wound inflammation in the longer-term observation period. The results from specific horn site left/right temperature measurements are only discussed in regard to the POS group and potential reasons for elevated temperatures at the horn sites for this specific group. Fig. 2 seems sufficient to communicate the key message to the reader, and Fig. 3 could be moved to the supplemental section.
Nociceptive threshold (Table 3, Fig. 4 and 5) – There does not seem a lot of value in analyzing and depicting the same data 3 different ways to later determine that ‘the lack of significant difference between the disbudded treatments stems from a higher level of individual calf MNT variability’. This should be consolidated to one figure or table, with others left out or moved to a supplemental section.”
Response: Suggestion actioned. Many tables and figures have been relegated to Supplementary Materials or removed for a more concise and hopefully more readable results section.
With regard to your specific feedback, we removed two of the MNT visualisations. However, we did keep the original Figure 3 as the difference in horn sites is something prominently discussed in terms of sun exposure etc.
Comment: “The simple summary and abstract, which draw in the reader’s attention for a publication both mention that ‘pain and discomfort/ symptoms associated with disbudding is/are suggested to last up to 62 days’, which immediately raises the question why the study was designed for a 7-day treatment and 12-day observation period of calves with analgesics post-disbudding, and not for longer? Suggestion to re-phrase the statement and/or discuss the study design choices. In the introduction it is mentioned that ‘Emerging research reveals that impaired welfare status persists throughout the tissue repair process, which can total up to 62 days’, which gives some more context to the statement from before, but could still raise confusion about the study design. I was unable to verify the source for this (Ref. 15 is given, which is ‘ Meijboom, F.; Stafleu, F. Farming ethics in practice: from freedom to professional moral autonomy for farmers. Agriculture 938 and Human Values 2015, 33, 403-414., which did not contain information on the 62 day statement as far as I could see).”
Response: We have changed that part of the abstract, summary and intro to two weeks, with the correct reference of Adcock & Tucker, 2018. The choice for 7-day feeding period and 12-day experimental period is largely a matter of resources, and the fact that this trial is a ‘proof of concept’ for sustained analgesic treatment. We make it clear that the study is looking longer-term than the single-dose meloxicam injection but perhaps not as long-term as it could be. Study design choice is also discussed at 4.7.
Comment: “All figure legends would benefit from the addition of group sizes to the legend.”
Response: Actioned.
Comment: “1 – Would benefit from adding the effective therapeutic plasma concentration of meloxicam in calves as a horizontal line”
Response: We agree that such a line, and threshold would be valuable. However, there is no exact consensus on effective concentration for calves (and extrapolation from other species not recommended). See Jokela et al., 2023.
Comment: “Section 4.1 – It should be stated if the meloxicam plasma concentrations represented the effective dose for calves on all days of testing.”
Response: With reference to Jokela et al., 2023, I have added some discussion of the limitation of not having a clear effective plasma concentration for calves.
Comment: “Section 4.2 – The choice of IR temperature as endpoint/method should be critically discussed for unexpected temperature measurements. The method is known to only provide information on surface temperatures, which can be affected by outside influences, as you discussed. It is also recommended to take at least 3 separate measurements and average them for one timepoint, which I believe wasn’t done here.”
Response: Discussion point added.
Comment: “Section 4.6 – Another point that could be added here is that provision of analgesics in form of a medicated diet reduces animal handling stress and prevents needle-shyness in animals, thus improving animal welfare. This becomes especially relevant if provision of analgesia beyond the first 48h after disbudding is pursued.”
Response: We have also had that thought. Added an extra line in 4.7 Future Directions!
Comment: “At least in the version I am reviewing, the reference section appears to be misaligned – some references seem to be formatted as sub-references, thereby creating number repetitions. Please review and update if found erroneous on your end.”
Response: Formatting error. Actioned.
Comment: “Section 2.4.1 – Should state the frequency of blood collections”
Response: Actioned.
Comment: Section 2.4.2 – Text reference to ethogram/Table 1 should be added
Response: Actioned.
Comment: Line 48 – Add comma after ‘hyperalgesia’
Response: Actioned.
Comment: Line 288 – Repeat word ‘that’ can be deleted
Response: Actioned.
Reviewer 2 Report
Comments and Suggestions for Authors
Dear Authors,
this manuscript addresses a topic of great relevance to animal welfare, proposing an innovative strategy for pain management in disbudded dairy calves. The use of medicated meloxicam pellets as an alternative to the conventional injectable formulation is practical and promising. Although the study presents interesting results, there are some methodological issues and interpretative gaps that should be addressed to strengthen the reliability and clarity of the conclusions.
Lines 54-55 & 751-769: The study period extends up to the 12th day after disbudding, while literature suggests that pain and tissue sensitivity may persist up to 62 days. Clarify this aspect in the limitations of the study and in the future perspectives to evaluate the effects of long-term treatment.
Line 116: The MMP group received meloxicam through pellet consumption, without individualized intake monitoring. This introduces a significant source of variability, as the actual dose consumed per animal is unknown. Includes some line on how intake variability may have affected the treatment's efficacy.
Lines 108-121: Each treatment group consisted of only 10 calves. This sample size limits statistical power, especially for results with high individual variability (e.g. nociceptive threshold). I suggest adding a few lines on statistical limitations in the discussion part.
Line 172: Correct "dependent on calf visibility" with "depending on calf visibility"
Table 1 fourth line: Correct "Rapid alternately lifting" with "Rapid alternate lifting"
Line 160: Correct with "anticoagulant"
Line 185: Change "sufficient memory for capturing data" with "sufficient storage capacity".
Line 186: Correct "proportion of each day that the calves spent lying" with "proportion of the day calves spent lying down"
Line 189: Correct "time-matched to accelerometer data" with "synchronized with accelerometer data".
Lines 217-260: The construction of the Behavioral Score (BS) raises some concerns. All behaviors are given equal weight without justification and no validation of the score is provided against independent measures of pain. Clarify the rationale for the inclusion and weighting of specific behaviors.
Lines 454-474: Despite observed differences in horn site inflammation and behavior, the mechanical nociceptive threshold (MNT) did not significantly differ between disbudded groups after Day 2. Do you have an explanation for this discrepancy? Could MNT be less sensitive or influenced by high inter-individual variation? Add it in the discussion part.
Line 585: Correct "as classed as associated with pain" with "classified as associated with pain" or "considered associated with pain"
Lines 638-644: Lines Horn site temperatures were likely affected by sunlight exposure, particularly in the POS group. Since measurements were conducted in an unshaded area, this could compromise the validity of temperature comparisons. Address the limitations of IR temperature collection in the discussion part, and add some suggestion (shading or standardization, etc.) for future protocols.
Author Response
Thank you for your feedback.
Comment: “Lines 54-55 & 751-769: The study period extends up to the 12th day after disbudding, while literature suggests that pain and tissue sensitivity may persist up to 62 days. Clarify this aspect in the limitations of the study and in the future perspectives to evaluate the effects of long-term treatment.”
Response: 62 days part removed, now two weeks. Clarified in introduction and discussed at 4.6.
Comment: “Line 116: The MMP group received meloxicam through pellet consumption, without individualized intake monitoring. This introduces a significant source of variability, as the actual dose consumed per animal is unknown. Includes some line on how intake variability may have affected the treatment's efficacy.”
Response: Actioned. See 4.5.
Comment: “Lines 108-121: Each treatment group consisted of only 10 calves. This sample size limits statistical power, especially for results with high individual variability (e.g. nociceptive threshold). I suggest adding a few lines on statistical limitations in the discussion part.”
Response: Actioned. See 4.5.
Comment: “Line 172: Correct "dependent on calf visibility" with "depending on calf visibility"”
Response: Actioned.
Comment: Table 1 fourth line: Correct "Rapid alternately lifting" with "Rapid alternate lifting"
Response: Actioned.
Comment: Line 160: Correct with "anticoagulant"
Response: Actioned.
Comment: “Line 185: Change "sufficient memory for capturing data" with "sufficient storage capacity".”
Response: Actioned.
Comment: “Line 186: Correct "proportion of each day that the calves spent lying" with "proportion of the day calves spent lying down"”
Response: Actioned.
Comment: “Line 189: Correct "time-matched to accelerometer data" with "synchronized with accelerometer data".”
Response: Actioned.
Comment: “Lines 217-260: The construction of the Behavioral Score (BS) raises some concerns. All behaviors are given equal weight without justification and no validation of the score is provided against independent measures of pain. Clarify the rationale for the inclusion and weighting of specific behaviours.”
Response: Each behaviour was given equal weight because there is no evidence to suggest that certain behaviours are more or less indicative of pain and discomfort than other behaviours, for all calves. A whole different study would have to be done to have solid support for giving different behaviours different statistical weights. Regardless, behavioural score removed for low value.
Comment: “Lines 454-474: Despite observed differences in horn site inflammation and behavior, the mechanical nociceptive threshold (MNT) did not significantly differ between disbudded groups after Day 2. Do you have an explanation for this discrepancy? Could MNT be less sensitive or influenced by high inter-individual variation? Add it in the discussion part.”
Response: See third paragraph in 4.2. Essentially, we needed more biological replicates for clearer trend. Also a potential indicator of distinction between inflammation and nociceptive pathways, as they are different.
Comment: “Line 585: Correct "as classed as associated with pain" with "classified as associated with pain" or "considered associated with pain"”
Response: Actioned.
Comment: “Lines 638-644: Horn site temperatures were likely affected by sunlight exposure, particularly in the POS group. Since measurements were conducted in an unshaded area, this could compromise the validity of temperature comparisons. Address the limitations of IR temperature collection in the discussion part, and add some suggestion (shading or standardization, etc.) for future protocols.”
Response: Discussed at length in 4.2.
Reviewer 3 Report
Comments and Suggestions for Authors
I would like to thank the authors for this comprehensive study to avoid pain of disbudding in heifers to enhance their welfare.
Abstract - satisfy
Introduction- Well written with appropriate references.
Methodology-
Line - 119-121- How would you assess the total amount of the MMP consumption of a heifer if you provide access to a common feed trough after 2 hrs? This is important to assess the consumed drug dosage for each heifer.
Line 129-130- Did you practice the restraint protocol for the sham disbudding group? Because restraining also causes stress and can be affected on the behaviour parameters.
Better to explain how did you decided the dose rate of 120mg/kg pellets?
Results
Line 177- The title of the table should come top of the ethogram.
Please define MMP, MET, POS, NEG in each figure caption
Discussion- satisfy
Conclusion- Satisfy
Author Response
Thank you for your feedback.
Comment: “Line - 119-121- How would you assess the total amount of the MMP consumption of a heifer if you provide access to a common feed trough after 2 hrs? This is important to assess the consumed drug dosage for each heifer.”
Response: We agree that it would have been ideal to be able to assess the exact pellet consumption of each MMP calf. However, some of the calves weren’t eating their pellets in their individual pens but were consuming pellets if put in their common trough. Since we did not have the manpower to be able to individually monitor all five pellet calves’ consumption from a common feed trough or leave them in the individual pens for such an extended period of time, the decision was made to give them the opportunity to consume the pellets for 2hrs and then put the remainder into the common feed trough. There is undoubtedly some variability in consumption, as discussed in 4.5, however we believe that would happen with any set of calves so the data is still an apt representation of on-farm results.
Comment: “Line 129-130- Did you practice the restraint protocol for the sham disbudding group? Because restraining also causes stress and can be affected on the behaviour parameters.”
Response: The sham disbudding group were sedated for their sham-disbudding procedure and so no restraint was needed. During data collection, they were restrained the same as all other calves.
Comment: “Better to explain how did you decided the dose rate of 120mg/kg pellets?”
Response: Rate decision is clarified in the paper – it was calculated by previous studies done by the research group.
Comment: “Line 177- The title of the table should come top of the ethogram.”
Response: Actioned.
Comment: “Please define MMP, MET, POS, NEG in each figure caption”
Response: Actioned.
Round 2
Reviewer 2 Report
Comments and Suggestions for Authors
Dear Authors,
I would like to sincerely congratulate you on the quality of your work. The manuscript is clearly written, methodologically rigorous, and represents a valuable contribution to the field.
Best Regards